# Surface Shortwave Net Radiation Estimation from Landsat TM/ETM+ Data Using Four Machine Learning Algorithms

**Yezhe Wang** [1,2]**, Bo Jiang** [1,2,]*****, Shunlin Liang** [3]**, Dongdong Wang** [3]**, Tao He** [4]**, Qian Wang** [2]**, Xiang Zhao** [1,2] **and Jianglei Xu** [1,2]

[1] State Key Laboratory of Remote Sensing Science, Jointly Sponsored by Beijing Normal University and Institute of Remote Sensing and Digital Earth of Chinese Academy of Sciences, Beijing 100875, China; wangyz@mail.bnu.edu.cn (Y.W.); zhaoxiang@bnu.edu.cn (X.Z.); jiangleixu@mail.bnu.edu.cn (J.X.)

[2] Beijing Engineering Research Center for Global Land Remote Sensing Products, Faculty of Geographical Science, Institute of Remote Sensing Science and Engineering, Beijing Normal University, Beijing 100875, China; qianwang@bnu.edu.cn

[3] Department of Geographical Sciences, University of Maryland at College Park, College Park, MD 20742, USA; sliang@umd.edu (S.L.); ddwang@umd.edu (D.W.)

[4] School of Remote Sensing and Information Engineering, Wuhan University, Wuhan 430079, China; taohers@whu.edu.cn

***** Correspondence: bojiang@bnu.edu.cn; Tel.: +86-10-58804250

**Abstract:** Surface shortwave net radiation (SSNR) flux is essential for the determination of the radiation energy balance between the atmosphere and the Earth's surface. The satellite-derived intermediate SSNR data are strongly needed to bridge the gap between existing coarse-resolution SSNR products and point-based measurements. In this study, four different machine learning (ML) algorithms were tested to estimate the SSNR from the Landsat Thematic Mapper (TM)/ Enhanced Thematic Mapper Plus (ETM+) top-of-atmosphere (TOA) reflectance and other ancillary information (i.e., clearness index, water vapor) at instantaneous and daily scales under all sky conditions. The four ML algorithms include the multivariate adaptive regression splines (MARS), backpropagation neural network (BPNN), support vector regression (SVR), and gradient boosting regression tree (GBRT). Collected in-situ measurements were used to train the global model (using all data) and the conditional models (in which all data were divided into subsets and the models were fitted separately). The validation results indicated that the GBRT-based global model (GGM) performs the best at both the instantaneous and daily scales. For example, the GGM based on the TM data yielded a coefficient of determination value ($R^2$) of 0.88 and 0.94, an average root mean square error (RMSE) of 73.23 W·m$^{-2}$ (15.09%) and 18.76 W·m$^{-2}$ (11.2%), and a bias of 0.64 W·m$^{-2}$ and –1.74 W·m$^{-2}$ for instantaneous and daily SSNR, respectively. Compared to the Global LAnd Surface Satellite (GLASS) daily SSNR product, the daily TM-SSNR showed a very similar spatial distribution but with more details. Further analysis also demonstrated the robustness of the GGM for various land cover types, elevation, general atmospheric conditions, and seasons

**Keywords:** surface shortwave net radiation; Thematic Mapper (TM); Enhanced Thematic Mapper Plus (ETM+); Landsat; machine learning model; remote sensing

## 1. Introduction

Surface energy fluxes profoundly affect our ability to understand how Earth's climate responds to increasing concentrations of greenhouse gases [1]. Surface shortwave net radiation (SSNR) is defined

as the difference between the incoming and outgoing shortwave radiation fluxes (0.3–3 μm) at the ground level, and determines the surface radiative energy balance during the daytime [2,3]. Therefore, it plays an important role in most energy-consuming ecosystem procedures, for example, it is the dominant source of melting energy on most glaciers [4], and is relevant to the processes of crop evapotranspiration [5], as well as the temporal variation of land surface temperature [6].

The use of point-based shortwave radiation field measurements are believed to have higher accuracy than other sources, but limited due to the low spatial density of measurements. Therefore, numerous studies have been devoted to the development of shortwave radiation estimation algorithm and products generation [7–14]. At present, the shortwave radiation algorithms can be separated into two categories: Model-based and satellite-based. Based on climate models' simulations, a gap of more than 10 W·m$^{-2}$ in global annual mean SSNR with 10 km ≥ pixel scale still existed [15]. Nowadays, satellite observations have become a key source for surface energy budget component estimation because of its unique advantages [16]. Several algorithms have been successfully developed based on various satellite data, such as geostationary satellite data (i.e., the Geostationary Operational Environmental Satellite, GOES) [13,14,17], polar-orbiting satellite data (i.e., Moderate Resolution Imaging Spectroradiometer (MODIS)) [18,19], and even multi-source remotely sensed data [20]. However, all of the available shortwave radiation estimates or products have medium-to-coarse spatial resolution. Even the latest product with the highest spatial resolution named GLASS (Global Land Surface Satellite) still has a 5 km scale (0.05°) [21,22]. This means that the existing products are unable to provide the information, which would enable the heterogeneous terrestrial features and fine-scale radiation budget changes that result from natural or anthropogenic processes, such as wildfire, urbanization, or agriculture [23–26] to be characterized. Besides, the accuracy of radiative budgets from various products has not been validated comprehensively because of the lack of SSNR data at a finer spatial resolution [27,28]; thus, their conclusions were somewhat incomplete. Therefore, SSNR data at an intermediate spatial resolution with high quality are urgently needed.

Landsat 30-m satellite data represent the only record of global land surface conditions at a spatial scale of 30 m spanning the last 40 years [29], with time-series Landsat data readily available. Consequently, the Landsat5/TM (Thematic Mapper) and Landsat7/ETM+ (Enhanced Thematic Mapper Plus) data have become the most common data used for the estimation of surface radiation budget [30–32]. Liang et al. [16] indicated that establishing the relationship between the top-of-atmosphere (TOA) radiance and surface incident insolation based on extensive radiative transfer simulations is a choice for calculating insolation and also inspired the estimation of SSNR [12,18,32,33]. Recently, Wang [32] developed a modified hybrid model (WangLUT, hereinafter) to estimate SSNR from Landsat5/TM and Landsat7/ETM+ TOA data. This model is based on the study conducted by Kim et al. [18] for retrieving SSNR using Moderate Resolution Imaging Spectroradiometer (MODIS) TOA and surface reflectance data. In the WangLUT method, the radiative transfer model MODTRAN5 is used firstly to establish the simulations of SSNR and TOA spectral radiance under various atmospheric and surface conditions base on Landsat data. Then, the regression relationships between the simulated instantaneous SSNR and TOA reflectance are pre-calculated and stored in a look-up table (LUT) by taking into account the cloud coverage and water vapor. Hence, the instantaneous SSNR can be calculated via the use of LUT. Lastly, the daily SSNR is calculated from the integral of the instantaneous SSNR. The observations from six AmeriFlux sites were used for validation, and the results showed an RMSE of 77.5 W·m$^{-2}$ and 36.1 W·m$^{-2}$ for the instantaneous and daily SSNR, respectively, which demonstrated the performance of this method. However, the limitations of the WangLUT method cannot be ignored. First of all, the LUT is built based on the MODTRAN5 simulations, which is suitable for large size experimental data, but is not good enough to characterize the complicated land surface and atmosphere conditions, such as topography, cloud shadow, and the case where the amount of water vapor (<0.5 g·cm$^{-2}$) is extremely low. Secondly, the LUT was determined based on the multivariate linear regression model, whose robustness still requires further verifications because the validation samples used were only within the United States (U.S.) territory.

In this study, several improvements are brought to address these issues. Firstly, comprehensive in-situ measurements were collected and used for both the modeling and validation. Then, new empirical models were developed with machine learning algorithms to replace the LUT of the WangLUT method, separately for the estimation of SSNR at instantaneous and daily scales. Finally, the new models were fully evaluated and analyzed. This paper is then organized as follows: Section 2 introduces the data and methods, whereas the validation and analysis results are provided in Section 3, and the conclusions are given in Section 4.

## 2. Data and Methods

### 2.1. Data

The data used in this study include in-situ measurements, remotely sensed data, reanalysis data, and other parameters. Table 1 shows the variables of the new model developed in this study, as well as their sources. More details are also provided in the next chapters.

**Table 1.** Variables of the surface shortwave net radiation (SSNR) model and their sources.

|  | Abbr. | Name | Unit | Temporal Resolution | Source |
|---|---|---|---|---|---|
| Response variable | SSNR | Surface shortwave net radiation | $W \cdot m^{-2}$ | instantaneous/daily | In-situ |
| Independent variables | SZA | Solar zenith angle | degree | instantaneous | Landsat data |
|  | SAA | Solar azimuth angle | degree | instantaneous | Landsat data |
|  | WV | Water vapor | $g \cdot cm^{-2}$ | hourly/daily | MERRA-2 |
|  | LAT | Latitude | degree | - | In-situ |
|  | $r_i{}^1$ | Top-of-atmosphere (TOA) reflectance | \ | instantaneous | Landsat data |
|  | BT | Brightness temperature | K | instantaneous | Landsat data |
|  | $CI^2$ | Clearness index | \ | daily | Calculated |

[1] $r_i$ ($i$ = 1, 2, 3, 4, 5, 7) is the $i$ th band TOA reflectance. [2] CI is used only for daily SSNR estimation.

### 2.1.1. In-Situ Measurements

Mathematically, SSNR can be expressed as:

$$\text{SSNR} = R_g - R_{gout} = (1 - \alpha)R_g \tag{1}$$

where $R_g$ and $R_{gout}$ are the downwelling and upwelling shortwave radiation fluxes at the surface, and $\alpha$ is the surface broadband shortwave albedo. The SSNR measurements were calculated from the observed $R_g$ and $R_{gout}$ and collected from 171 sites belonging to 10 global measurement networks during 1994–2011. Table 2 provides detailed information on each network.

Figure 1 shows the geographical distributions of all sites and the climate zones they belong to. These climate zones are defined following the Köppen–Geiger climate classification [48]. These sites are located across the globe and therefore represent the different climatic and ecosystem conditions, ranging from the Arctic to the Antarctic. Figure 2 shows the elevations of all sites ranging from 1 to 3423 m above sea level, and Table 3 provides the statistics of the land cover types of these sites defined by the International Geosphere–Biosphere Programme (IGBP). In this paper, 11 land cover types were considered, including evergreen needleleaf forest (ENF), evergreen broadleaf forest (EBF), deciduous needleleaf forest (DNF), deciduous broadleaf forest (DBF), mixed forest (MF), shrublands (SHB), savannas (SAV), grasslands (GRA), permanent wetlands (PW), croplands (CRO), and barren lands (BR). Note that some land cover types (i.e., ice, snow and water body) were not considered due to the lack of available samples. The comprehensive representation of land cover types and climate

types, widespread spatial distribution, and different elevations allows the global applicability of the model to be developed and evaluated.

**Table 2.** Information related to the 10 measurement networks.

| Abbreviations | Time Period | Instrument | Reference | Temporal Resolution |
|---|---|---|---|---|
| ARM[1] | 1994–2011 | Kipp&Zonen Pyrgeometer | [34] | 1 min |
| AsiaFlux | 2000–2009 | Kipp&Zonen, CM-6F | [35] | 30 min |
| BSRN[2] | 1995–2011 | Eppley, PIR/Kipp&Zonen CG4 | [36] | 1 or 3 min |
| CEOP[3] | 2008–2009 | - | - | 30 min |
| EOL[4] | 2006–2007 | Kipp&Zonen CM21, Kipp&Zonen CG4s | [37] | 1 h |
| GC_NET[5] | 1997–1998 | Li Cor Photodiode & REBS Q* 7 | [38] | 1 h |
| La Thuile[6] | 1997–2011 | Kipp&Zonen Pyrgeometer, etc | [39] | 30 min |
| SAFARI.2000[7] | 2000 | Kipp&Zonen Pyrgeometer | [40,41] | 30 min |
| SURFRAD[8] | 1995–2011 | Eppley, PIR | [42] | 3 min |

[1] ARM: Atmospheric Radiation Measurement. [2] BSRN: Baseline Surface Radiation Network [43]. [3] CEOP: Coordinated Enhanced Observation Network of China [44,45]. [4] EOL: Earth Observing Laboratory. [5] GC_NET: Greenland Climate Network [46]. [6] La Thuile: Global Fluxnet (La Thuile dataset). [7] SAFARI.2000: Southern African Regional Science Initiative Project. [8] SURFRAD: Surface Radiation Network [47].

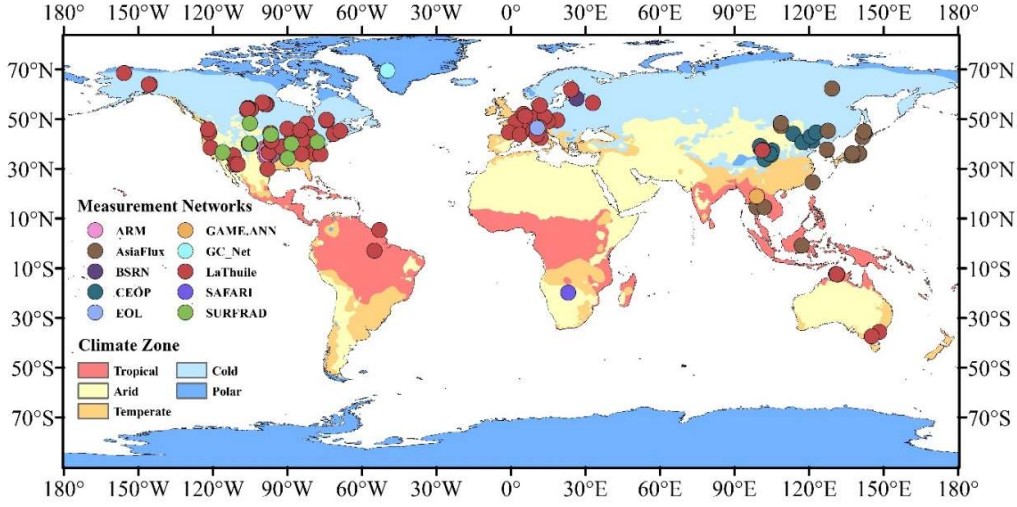

**Figure 1.** Distribution of 171 sites among 10 measurement networks and climate zones.

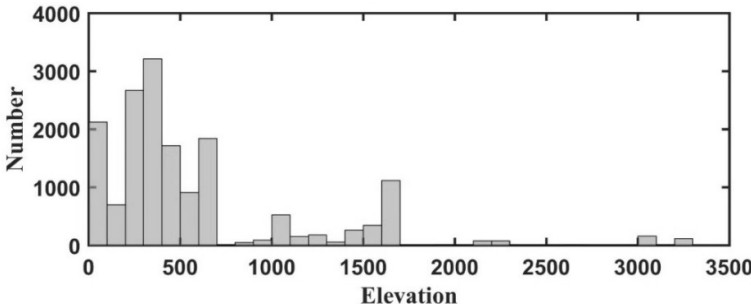

**Figure 2.** Histogram of the distribution of all observing sites elevation.

**Table 3.** Number of samples for each International Geosphere–Biosphere Programme (IGBP) land cover type.

| Main Types | Total Number of Samples | |
|---|---|---|
| | Instantaneous SSNR | Daily SSNR |
| ENF[1] | 2246 | 2165 |
| EBF[2] | 127 | 123 |
| DNF[3] | 90 | 97 |
| DBF[4] | 1072 | 1004 |
| MF[5] | 488 | 552 |
| PW[6] | 55 | 61 |
| CRO[7] | 3311 | 2619 |
| ICE[8] | 9 | - |
| BR[9] | 512 | 228 |
| SHB[10] | 411 | 405 |
| SAV[11] | 269 | 271 |
| GRA[12] | 7794 | 4629 |
| Total | 16,384 | 12,154 |

[1] ENF: Evergreen Needleleaf Forest. [2] EBF: Evergreen Broadleaf Forest. [3] DNF: Deciduous Needleleaf Forest. [4] DBF: Deciduous Broadleaf Forest. [5] MF: Mixed Forest. [6] PW: Permanent Wetlands. [7] CRO: Croplands. [8] ICE: Ice and snow. [9] BR: Barren Lands. [10] SHB: Evergreen Broadleaf Forest. [11] SAV: Savannas. [12] GRA: Grasslands.

The radiation measurements were all pre-processed according to the requirements of this study. After strict quality control [49], the timing of these observations was first converted into solar time for consistency, following which the radiation measurements ($R_g$ and $R_{gout}$) were aggregated into hourly (transit time being taken as the midpoint) and diurnal mean values. In this study, "instantaneous" is defined as the hourly scale, considering the various sampling time of each site (see Table 2). Lastly, the instantaneous and daily SSNR observations were calculated for each site and their values are shown in Table 3. To ensure the quality of all estimates, daily values were only calculated if at least one observation was available in each hour during a diurnal period, and any unreasonable values were manually removed at last.

### 2.1.2. Remotely Sensed Data

The TOA reflectance data from the Landsat 5/TM and 7/ETM+ were used as inputs for the new model. For comparison purposes, the GLASS (Global LAnd Surface Satellite) SSNR products (calculated from downwelling solar radiation and surface albedo) were also used.

● Landsat 5/TM and 7/ETM+ data

Landsat project is a joint initiative between the U.S. Geological Survey (USGS) and NASA. With a time coverage dating back to the 1980s, the Landsat multispectral data represent the world's longest continuously acquired collection of space-based land data, and has a relatively high resolution (~1 km) compared with that of other remote sensing projects. In this study, the data from Landsat 5/TM were used for modeling and the Landsat 7/ETM+ data were used for the evaluation of the model applicability. The data from the TM and ETM+ sensors onboard the Landsat 5 and 7 satellite have been proven extremely useful for various field studies such as land surface radiation budget [30,50,51], surface temperature [52], or surface albedo [53] estimation. The characteristics of all bands in Landsat 5/TM and Landsat 7/ETM+ are shown in Table 4, demonstrating that the bands properties from the two sensors are nearly the same for bands 1–7. Therefore, the data from the two sensors are used together in this study from 1994 to 2011.

**Table 4.** Band settings in Landsat 5/TM (Thematic Mapper) and Landsat 7/ETM+ (Enhanced Thematic Mapper Plus) sensors.

| Landsat 5/TM | | | Landsat 7/ETM+ | | |
|---|---|---|---|---|---|
| Band | Wavelength | Resolution | Band | Wavelength | Resolution |
| 1 | 0.45–0.52 | 30 m | 1 | 0.45–0.52 | 30 m |
| 2 | 0.52–0.60 | 30 m | 2 | 0.52–0.60 | 30 m |
| 3 | 0.63–0.69 | 30 m | 3 | 0.63–0.69 | 30 m |
| 4 | 0.76–0.90 | 30 m | 4 | 0.77–0.90 | 30 m |
| 5 | 1.55–1.75 | 30 m | 5 | 1.55–1.75 | 30 m |
| 6 | 10.40–12.50 | 120 m | 6 | 10.40–12.50 | 60 m |
| 7 | 2.08–2.35 | 30 m | 7 | 2.08–2.35 | 30 m |
| | | | 8 | 0.520–0.900 | 15 m |

After geometric and radiometric calibration, the TOA reflectance (band 1–5, and 7), BT (band 6), geometry information (SZA and SAA), and transit time of Landsat 5/TM and 7/ETM+ could be obtained directly. Then, the pixels corresponding to each site location were extracted in all available TM or ETM+ scenes. In the data pre-processing procedure, the LEDAPS (Landsat Ecosystem Disturbance Adaptive Processing System) atmosphere correction tool (version 3.3) [54,55] was applied and the generated cloud mask (using a method for cloud and cloud shadow detection, called Fmask (function of mask) [56])) was also used to classify the sky conditions, such as cloudy (qc values $\in$ [2,34]), cloud shadow (qc values $\in$ [4,12,20,36,52]), and clear (qc values $\in$ others), in this study. As mentioned by Zhu et al. [56], the overall cloud accuracy of Fmask was as high as 96.4%, and even the worst accuracy for cloud shadow was still more than 70%. The Fmask has been used frequently in previous studies [57]. Note that the saturated pixels whose TOA values in some bands are abnormal due to highly reflective surfaces (i.e., snow or clouds) were excluded [58,59].

• GLASS products

The GLASS satellite products of daily downwelling solar radiation (DSR) [3,21] and daily shortwave broadband albedo [3,60] at 5 km spatial resolution were used in this study to obtain the daily SSNR, according to Equation (1), for comparison purpose. GLASS DSR product was retrieved from multiple polar-orbiting and geostationary satellite data by LUT. GLASS's albedo product is produced from the advanced very high-resolution radiometer (AVHRR) and MODIS data based on different estimation algorithms. The two GLASS products have previously been evaluated using ground measurements and compared with other products [61,62], and the results of this comparison demonstrated that their accuracy is satisfactory, and even exceeded that of most of the other products. The comparison period was from 2000 and 2011 due to the limitation of the available GLASS data. The GLASS DSR and albedo products are available at: http://glass-product.bnu.edu.cn/.

### 2.1.3. MERRA-2 Reanalysis Data

The Modern-Era Retrospective Analysis for Research and Applications, version 2 (MERRA-2), is a global atmospheric reanalysis dataset produced by the NASA Global Modeling and Assimilation Office (GMAO) [63] at 0.5 x 0.625° spatial resolution. MERRA-2 is the updated version of MERRA [64], with improved estimates for various parameters, especially those related to the hydrological cycle. The hourly total precipitable water vapor (WV, g/cm$^2$) from MERRA-2 M2T1NXSLV [65] was extracted for each site and aggregated into average daily values.

### 2.1.4. Other Parameters

In this study, CI, which represents the daily atmospheric transmittance [66], was used as an input. It is defined as the ratio of the daily solar radiation and the extraterrestrial radiation as follows:

$$CI = \frac{R_g}{R_{se}} \tag{2}$$

where $R_g$(W·m$^{-2}$) is from the GLASS DSR product, and $R_{se}$(W·m$^{-2}$) is the extraterrestrial radiation calculated using Equation (3) from [67]:

$$R_{se} = \frac{1440 G_{sc} d_r}{\pi} (\omega_s \sin(\varphi) \sin(\delta) + \cos(\delta) \sin(\omega_s)) \tag{3}$$

$$d_r = 1 + 0.033 \cos\left(\frac{2\pi DOY}{365}\right) \tag{3a}$$

$$\delta = 0.409 \sin\left(\frac{2\pi DOY}{365} - 1.39\right) \tag{3b}$$

$$\omega_s = \arccos(-\tan(\varphi)\tan(\delta)) \tag{3c}$$

$$\text{rad} = \frac{\pi}{180} * (decimal\ deg) \tag{3d}$$

where $G_{sc}$ is the solar constant (0.0820 MJm$^{-2}$·min$^{-1}$),$d_r$ the inverse relative distance from the Earth to the Sun, $\omega_s$ the sunset hour angle (rad), $\varphi$ the latitude (rad), $\delta$ the solar declination (rad), z the elevation (m), and DOY is the day of the year.

### 2.2. Methods

The flowchart delineating the various processing for the implementation of the method described in this study is shown in Figure 3. After collection of the input data, the data were first randomly separating into two parts: 80% (13,116 samples for instantaneous and 9722 for daily) for training and the other 20% (3268 for instantaneous and 2432 for daily) for independent validation. Following this, four machine learning algorithms, including the multivariate adaptive regression splines (MARS) [68], the backpropagation neural network (BPNN) [69], the support vector regression (SVR) [70], and the gradient boosting regression tree (GBRT) [71], were used for the instantaneous SSNR (Ins-SSNR) and the daily SSNR (D-SSNR) estimation in the global mode (use of all data to fit the model) and conditional mode (classification of the data into clear, cloudy, and cloud shadows for instantaneous scale, and by three intervals of CI for daily scale, with the models being fitted separately), respectively. Thirdly, the optimal algorithm was determined for the Ins- and D-SSNR estimation after a comprehensive assessment. To be more objective, the performances of the new developed models were compared with the results from the WangLUT method, and more analytical experiments were also implemented to illustrate their robustness. Finally, the generated D-SSNR from Landsat 5/TM data was compared with the GLASS product. In this study, all the algorithms were implemented under the Microsoft Windows 10 system on an Intel Core 3.40 GHz PC with 8 GB memory. Details of the four algorithms are provided in Appendix A.

A 2-fold cross validation was conducted for the validation of the training accuracy, and four statistic index were used: The determination coefficient ($R^2$), root mean square error (RMSE), mean absolute error (MAE), and bias:

$$R^2 = 1 - \frac{\sum_{i=1}^{n}(e_i - o_i)^2}{\sum_{i=1}^{n}(o_i - \bar{o})^2} \tag{4a}$$

$$\text{RMSE} = \sqrt{\frac{1}{n}\sum_{i=1}^{n}(o_i - e_i)^2} \tag{4b}$$

$$\text{MAE} = \frac{1}{n}\sum_{i=1}^{n}|e_i - o_i| \tag{4c}$$

$$\text{bias} = \frac{1}{n}\sum_{j=1}^{n}(e_i - o_i) \tag{4d}$$

where $e_i$ is the estimated value, and $o_i$ is the ground observation for the $i$ th pair.

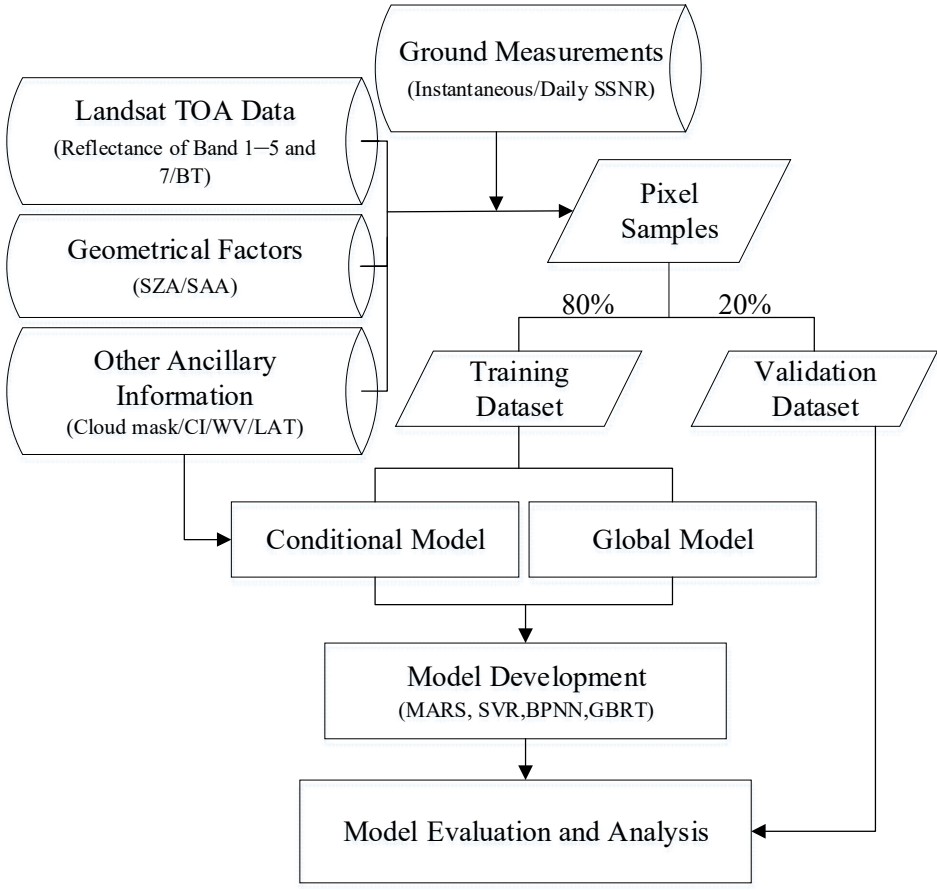

**Figure 3.** Flowchart of the methods used to estimate SSNR from Landsat data.

## 3. Results and Analysis

### 3.1. Model Development

After pre-processing, 16,384 instantaneous samples (13,116 for training and 3268 for validation) and 12,154 daily samples (9722 for training and 2432 for validation) were collected from 171 sites between 1994 and 2011 for Ins-SSNR and D-SSNR estimation models development and validation, respectively. The models (Ins-SSNR or D-SSNR) were fitted with the observations in two cases. In case 1, the models were fitted using all available observations, i.e., global model. In case 2, the observations were divided into several subsets based on the cloud mask (for Ins-SSNR) or CI (for D-SSNR), and the models thus obtained are referred to as conditional models.

#### 3.1.1. Ins-SSNR

The various effects of the atmosphere on the solar radiations (i.e., cloud, water vapor) need to be accounted for [32] to ensure accuracy of the estimates. For instance, in the WangLUT method, the SSNR estimates need to be corrected for the influence of water vapor. To highlight the difference with the WangLUT method, the BT and the WV were taken as input dimensions to estimate the Ins-SSNR (see

Table 1). As Equation (1) shows, the SSNR was determined by $R_g$ and surface shortwave broadband albedo, which was closely related to the land surface types and their change. Some previous studies demonstrated that Landsat BT data have been widely used for land cover classification and assessment of land use change [72,73], hence it was reasonable to be used as input to estimate the SSNR. The results with or without BT and/or WV are discussed later.

To ensure objectivity of the estimates, the model training accuracy was evaluated using a 2-fold cross-validation method with all Ins-SSNR training samples. Table 5 shows the training accuracy results of four Ins-SSNR estimation models in global mode. The results illustrate that the Ins-SSNR GBRT global model (GGM, hereinafter) performed the best, with a coefficient of determination $R^2$ of 0.86, an RMSE of 75.72 W·m$^{-2}$, an MAE of 51.51 W·m$^{-2}$ and a bias of –0.38 W·m$^{-2}$, when the BT and WV were both added as inputs. Thus, the GBRT algorithm had acceptable performances and a very short fitting time (0.03 sec), although the training time (2 hours) was a little longer.

As a global mode, Ins-SSNR conditional models under clear-, cloudy-, and cloud shadow-sky with different inputs were also developed. After several rounds of experiments, the GBRT algorithm was found to have superior performances to the other three algorithms in fitting efficiency and estimation accuracy. Hence, it was applied to develop the Ins-SSNR conditional models, and the results of the conditional model under these three different sky conditions are listed in Table 6. These results indicate that the Ins-SSNR GBRT conditional models (GCMs, hereinafter) performed the best under any condition when both the BT and WV were added as inputs. Relative to the cloud conditions, the accuracy of the model under cloud shadow is the worst, which is possibly due to the lower number of samples under this condition (887). It was also found that, with a higher RMSE (77.92 W·m$^{-2}$) and MAE (52.88 W·m$^{-2}$) the overall accuracy of the GCM, when combining the three conditional models results together (the last row in Table 6), is less good than that of the GGM (Table 5). Thus, the accuracy of the results from the GGM (with BT and WV as inputs) under the same three conditions was also examined, and is also displayed in Table 6. These results demonstrate that for most metrics, the performances in terms of accuracy of the GGM under the three conditions are better than those from the GCM. This is especially true in the case of cloud shadow condition with values of $R^2$ of 0.78 against 0.71, RMSEs of 90.89 against 102.35 W·m$^{-2}$, and MAE of 67.89 against 77.06 W·m$^{-2}$, respectively. Although the bias presents a less good result for the GGM with a value of 13.91 W·m$^{-2}$, the predicted results are closer to the 1:1 line and more concentrated. Therefore, it can be concluded that the GGM with BT and WV as inputs is the optimal model for Ins-SSNR estimation.

**Table 5.** 2-fold cross-validation results of the instantaneous SSNR (Ins-SSNR) global model using four algorithms with different inputs. RMSE, root mean square error; MAE, mean absolute error.

| | BT WV | | | | WV | | | | BT | | | | None | | | | Training Time | Fitting Time |
|---|---|---|---|---|---|---|---|---|---|---|---|---|---|---|---|---|---|---|
| | $R^2$ | RMSE (W·m⁻²) | MAE (W·m⁻²) | bias (W·m⁻²) | $R^2$ | RMSE (W·m⁻²) | MAE (W·m⁻²) | bias (W·m⁻²) | $R^2$ | RMSE (W·m⁻²) | MAE (W·m⁻²) | bias (W·m⁻²) | $R^2$ | RMSE (W·m⁻²) | MAE (W·m⁻²) | bias (W·m⁻²) | | |
| MARS | 0.84 | 83.39 | 58.06 | 0.05 | 0.84 | 83.44 | 58.17 | 0.18 | 0.83 | 85.46 | 60.04 | 0.17 | 0.83 | 85.46 | 60.04 | 0.17 | 1 min | 0.01 s |
| BPNN | 0.84 | 82.85 | 57.45 | 0.20 | 0.84 | 82.19 | 57.19 | 0.04 | 0.84 | 82.97 | 57.61 | 0.22 | 0.84 | 83.80 | 57.53 | −0.04 | 35 min | 0.06 s |
| SVR | 0.85 | 80.08 | 52.04 | 7.65 | 0.85 | 80.56 | 52.34 | 8.11 | 0.85 | 81.45 | 53.67 | 7.43 | 0.85 | 82.16 | 54.22 | 7.67 | 30 min | 1 s |
| GBRT | 0.86 | 75.72 | 51.51 | −0.38 | 0.86 | 76.88 | 52.25 | −0.11 | 0.86 | 77.47 | 52.92 | −1.05 | 0.86 | 78.15 | 53.49 | −0.91 | 2 h | 0.03 s |

**Table 6.** 2-fold cross-validation results of the Ins-SSNR gradient boosting regression tree (GBRT) global model (GGM) and GBRT conditional model (GCM) with different inputs

| | GGM | | | | | GCM | | | | | | | | | | | | | | | |
|---|---|---|---|---|---|---|---|---|---|---|---|---|---|---|---|---|---|---|---|---|---|
| | BT WV | | | | | BT WV | | | | WV | | | | BT | | | | None | | | |
| | Samples | $R^2$ | RMSE (W·m⁻²) | MAE (W·m⁻²) | bias (W·m⁻²) | $R^2$ | RMSE (W·m⁻²) | MAE (W·m⁻²) | bias (W·m⁻²) | $R^2$ | RMSE (W·m⁻²) | MAE (W·m⁻²) | bias (W·m⁻²) | $R^2$ | RMSE (W·m⁻²) | MAE (W·m⁻²) | bias (W·m⁻²) | $R^2$ | RMSE (W·m⁻²) | MAE (W·m⁻²) | bias (W·m⁻²) |
| Clear | 12,403 | 0.86 | 69.92 | 46.22 | −1.01 | 0.86 | 71.64 | 47.49 | 0.10 | 0.86 | 71.76 | 47.59 | −0.49 | 0.85 | 72.67 | 48.75 | −0.97 | 0.85 | 73.38 | 48.87 | −0.51 |
| Cloud | 3094 | 0.78 | 91.32 | 67.71 | −1.93 | 0.78 | 92.31 | 66.96 | −2.87 | 0.78 | 92.76 | 67.93 | −4.09 | 0.77 | 93.59 | 67.96 | −3.87 | 0.76 | 95.06 | 69.44 | −6.61 |
| Cloud shadow | 887 | 0.78 | 90.89 | 67.89 | 13.91 | 0.71 | 102.35 | 77.06 | 2.13 | 0.71 | 103.02 | 78.06 | −1.41 | 0.70 | 105.33 | 79.43 | −1.74 | 0.71 | 104.05 | 79.15 | −7.85 |
| Overall | 13,116 | 0.86 | 75.72 | 51.51 | −0.38 | 0.86 | 77.92 | 52.88 | −0.36 | 0.86 | 78.25 | 53.16 | −1.23 | 0.86 | 78.35 | 53.24 | −1.57 | 0.85 | 79.43 | 53.99 | −2.16 |

Besides, the contribution of each input in GGM for the Ins-SSNR estimation are displayed in Figure 4. The Ins-SSNR estimations are shown to be more sensitive to BT and WV than to most of the other bands, thus highlighting the importance of taking BT and WV as inputs.

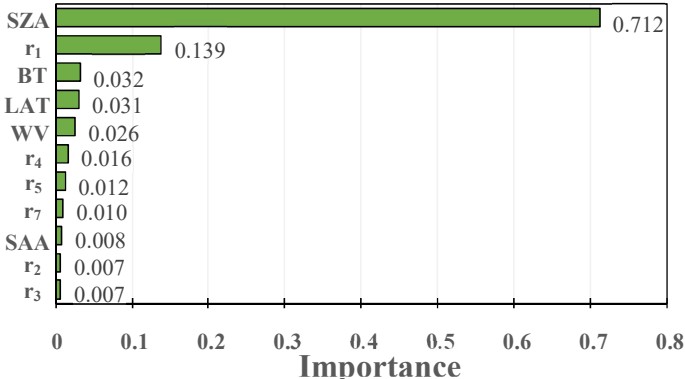

**Figure 4.** Contributions of each inputs in the GGM.

### 3.1.2. D-SSNR

In the WangLUT method, the D-SSNR was calculated from the integral of the Ins-SSNR based on the assumption that the atmospheric conditions were moderately invariant during the daytime. However, this assumption was found to be unreasonable after statistical analysis (see Figure 5). The cloud mask classification results were used as the atmospheric condition at the transit time (y axis), and the CI represents the daily average atmospheric condition (x axis). The results in this figure illustrate that no significant correlation exists between the two atmospheric conditions (instantaneous and daily) at the same date. To address this issue in this study, the CI was considered as one input to represent the daily average atmospheric condition. As in the Ins-SSNR, the D-SSNR estimation models were also developed using four algorithms with different inputs for both the global and conditional modes. However, in the D-SSNR case, the daily conditions are classified by CI, with three categories obtained: CI < 0.2 for overcast sky, 0.2 ≤ CI < 0.7 for cloudy sky, and CI ≥ 0.7 for clear sky. Following the results of the Ins-SSNR model testing, both BT and WV are taken as inputs in the D-SSNR modeling.

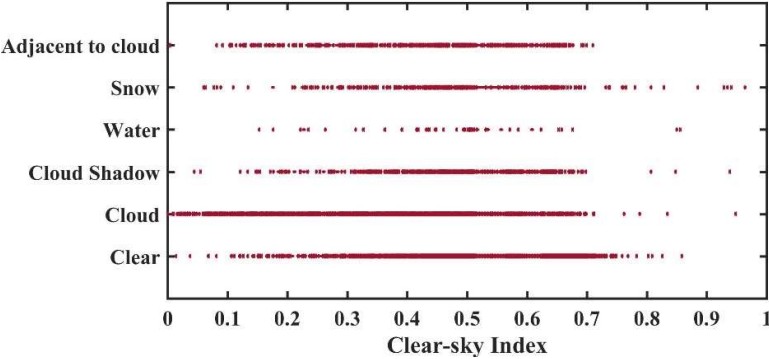

**Figure 5.** The statistical results of the relation between cloud mask and CI.

The 2-fold cross-validation results of D-SSNR estimation models using four algorithms in global mode with or without CI are summarized in Table 7. These results demonstrate that all models performed better when CI was added as an input. In this case too, the GGM exhibits the best performance with a $R^2$ of 0.93, an RMSE of 21.01 W·m$^{-2}$, an MAE of 14.07 W·m$^{-2}$, and a bias of 0.12 W·m$^{-2}$.

**Table 7.** 2-fold cross-validation results from D-SSNR estimation models using four algorithms in global mode with or without CI.

| | | CI | | | | Without CI | | | Training Time | Fitting Time |
|---|---|---|---|---|---|---|---|---|---|---|
| | $R^2$ | RMSE (W·m$^{-2}$) | MAE (W·m$^{-2}$) | bias (W·m$^{-2}$) | $R^2$ | RMSE (W·m$^{-2}$) | MAE (W·m$^{-2}$) | Bias (W·m$^{-2}$) | | |
| MARS | 0.91 | 24.10 | 15.30 | 0.04 | 0.86 | 29.13 | 20.50 | 0.19 | 1 min | 0.02 s |
| BPNN | 0.92 | 22.39 | 14.78 | −0.026 | 0.86 | 28.94 | 20.30 | 0.13 | 1.5 h | 0.03 s |
| SVR | 0.92 | 21.96 | 13.90 | 1.21 | 0.88 | 27.98 | 18.70 | 2.99 | 1.2 h | 0.80 s |
| GBRT | 0.93 | 21.01 | 14.07 | 0.12 | 0.88 | 27.21 | 18.94 | −0.35 | 2.5 h | 0.02 s |

The GBRT algorithm also works the best in conditional mode, thereby, only the results using GRBT algorithm are delineated here. Overall, the accuracy was better under all conditions when CI was added, especially for CI [0.2, 0.7), with a significant increased accuracy ($R^2$ of 0.91 and 0.87, RMSEs of 21.06 and 27.08 W·m$^{-2}$, MAEs of 14.55 and 19.42 W·m$^{-2}$, and biases of 0.24 and −2.53 W·m$^{-2}$, with and without CI as an input, respectively). However, the accuracy was the worst for CI < 0.2, which illustrates how thick clouds block much of the surface information. The overall accuracy from GGM (with CI) (Table 8) is similar to the combined results from the three GCM (with CI) (the last row in Table 9) with $R^2$ of 0.93 and 0.92, RMSEs of 21.01 and 21.34 W·m$^{-2}$, MAEs of 14.07 and 14.52 W·m$^{-2}$, and biases of 0.12 and 0.04 W·m$^{-2}$, respectively. The accuracy of the results from the D-SSNR GGM with CI under three conditions was also examined and is shown in Table 8. Similar conclusions as for the Ins-SSNR estimation are reached, i.e., that the GGM for D-SSNR estimations performs better than the GCM, particularly under the CI < 0.2 condition ($R^2$ of 0.63 and 0.57, RMSEs of 21.52 and 23.35 W·m$^{-2}$, MAEs of 11.70 and 12.38 W·m$^{-2}$, and bias of −0.22 and −3.16 W·m$^{-2}$ for the GGM and GCM, respectively).

**Table 8.** 2-fold cross-validation results from GGM and GCM with or without CI.

| | Samples | GGM | | | | GCM | | | | | | | |
| | | CI | | | | CI | | | | Without CI | | | |
| | | $R^2$ | RMSE (W·m$^{-2}$) | MAE (W·m$^{-2}$) | bias (W·m$^{-2}$) | $R^2$ | RMSE (W·m$^{-2}$) | MAE (W·m$^{-2}$) | bias (W·m$^{-2}$) | $R^2$ | RMSE (W·m$^{-2}$) | MAE (W·m$^{-2}$) | bias (W·m$^{-2}$) |
|---|---|---|---|---|---|---|---|---|---|---|---|---|---|
| CI < 0.2 | 414 | 0.63 | 21.52 | 11.70 | −0.22 | 0.57 | 23.35 | 12.38 | −3.16 | 0.56 | 23.77 | 12.52 | −2.94 |
| 0.2 ≤ CI < 0.7 | 9136 | 0.92 | 20.87 | 14.19 | 0.05 | 0.92 | 21.06 | 14.55 | 0.24 | 0.87 | 27.08 | 19.42 | −2.53 |
| CI ≥ 0.7 | 172 | 0.94 | 27.56 | 14.96 | −1.74 | 0.93 | 29.52 | 18.11 | 0.93 | 0.93 | 30.09 | 17.82 | −0.66 |
| Overall | 9722 | 0.93 | 21.01 | 14.07 | 0.12 | 0.92 | 21.34 | 14.52 | 0.04 | 0.88 | 26.97 | 19.08 | −2.51 |

**Table 9.** Samples for each elevation interval.

| Elevation Interval | Total Number of Samples | |
|---|---|---|
| | Instantaneous SSNR | Daily SSNR |
| <200 m | 582 | 370 |
| 200~400 m | 1157 | 978 |
| 400~600 m | 530 | 435 |
| 600~1000 m | 390 | 287 |
| 1000~1500 m | 242 | 156 |
| 1500~2000 m | 276 | 114 |
| 2000~3000 m | 40 | 33 |
| ≥3000 m | 51 | 59 |
| Total | 3268 | 2432 |

The results of the importance of each input in the GGM (Figure 6) also indicate that the addition of CI has a remarkable impact on D-SSNR estimation, thus highlighting the necessity of adding CI as an input. The use of CI was therefore chosen for the D-SSNR estimation model of the GGM.

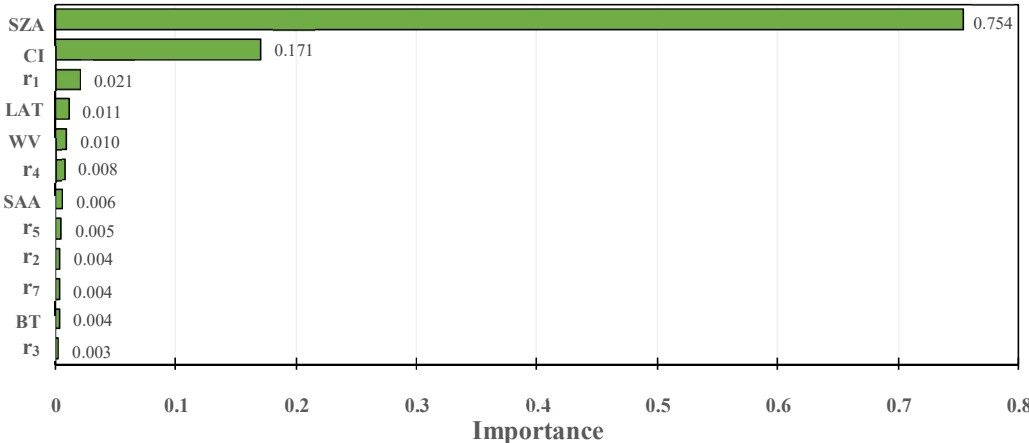

**Figure 6.** Contribution of each input in GGM.

## 3.2. Validation and Comparison

### 3.2.1. Direct Validation

The performance of the GGM for Ins-SSNR and D-SSNR estimates was evaluated with different inputs (BT and WV for Ins-SSNR, CI for D-SSNR), and the results were validated against independent validation samples (3268 for Ins-SSNR, 2432 for D-SSNR) as shown in Figure 7. For clarity purpose, the estimates under different conditions are represented by different colors, separately for Ins-SSNR (Figure 7a) and D-SSNR (Figure 7b). The Ins-SSNR estimates have an overall $R^2$ value of 0.88, RMSE of 73.23 W·m$^{-2}$, MAE of 49.19 W·m$^{-2}$, and bias of 0.64 W·m$^{-2}$, while the values for the D-SSNR estimates were 0.94, 18.76 W·m$^{-2}$, 13.06 W·m$^{-2}$, and –1.74 W·m$^{-2}$, respectively.

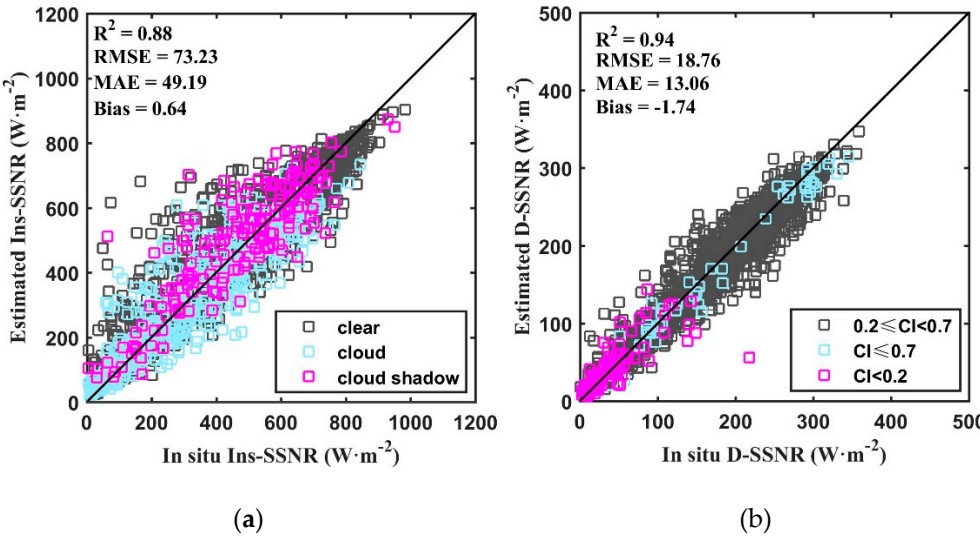

(**a**)          (**b**)

**Figure 7.** Independent validation results of Ins-SSNR (**a**) and daily SSNR (D-SSNR) (**b**) estimates obtained by the GGM.

The Ins-SSNR and D-SSNR results from WangLUT method were also validated for comparison, these results being shown in Figures 8 and 9. Note, however, that as the WangLUT method is only applicable under clear- and cloud-sky conditions, when WV is larger than 0.5 g·cm$^{-2}$, only the in-situ samples corresponding to these conditions (2901 for Ins-SSNR, 2010 for D-SSNR) were used for the validation.

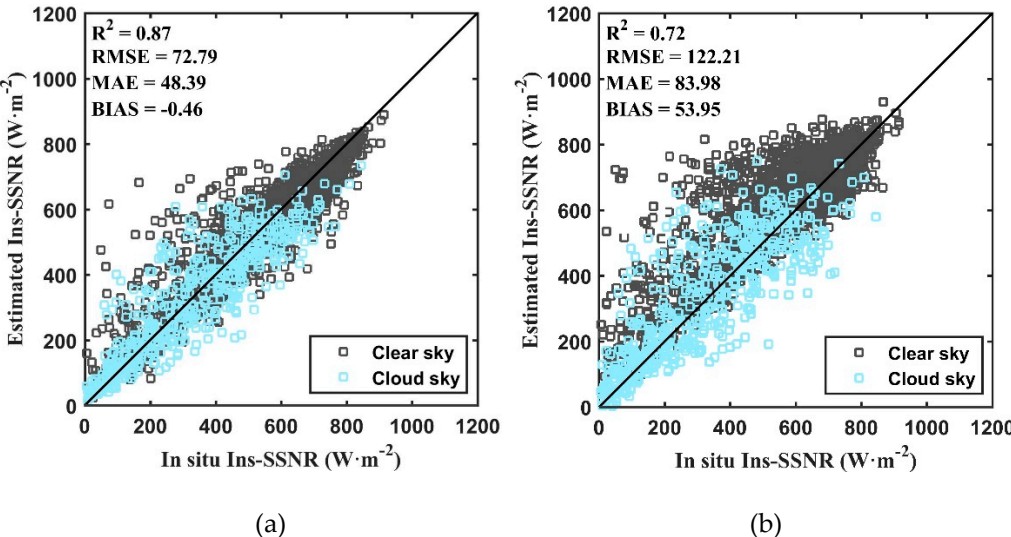

(a)                                                                                    (b)

**Figure 8.** Scatterplots of the comparison results of the Ins-SSNR estimates from (**a**) GGM, and (**b**) WangLUT method.

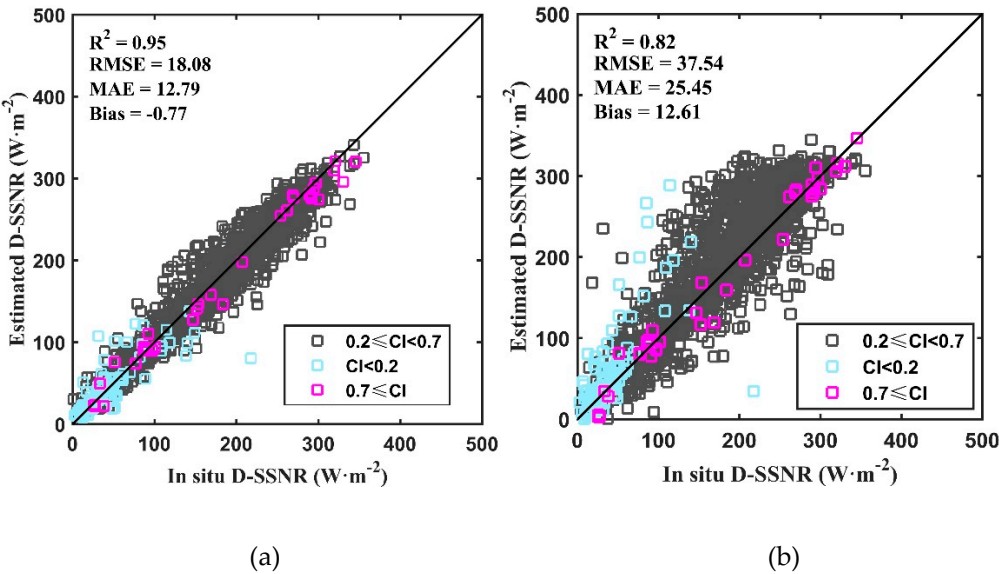

(a)                                                                                    (b)

**Figure 9.** Same as Figure 8, but for the D-SSNR.

Figure 8 demonstrates that the GGM-based estimates are much better than the ones from the WangLUT method, having a higher $R^2$ value (0.87) and lower RMSE (72.79 W·m$^{-2}$), MAE (48.39 W·m$^{-2}$), and bias values (–0.46 W·m$^{-2}$). Besides, the Ins-SSNR estimates under clear-sky are more dispersed in the WangLUT method, whereas the performances under cloud-sky are relatively similar for the two methods.

The validation accuracy of D-SSNR from the GGM is also better than the one from the WangLUT method (Figure 9), with an RMSE value of 18.08 W·m$^{-2}$ and bias of –0.77 W·m$^{-2}$, which is even better than the results of previous studies [32]. The D-SSNR estimates are also shown to be closer to the 1:1 line, though the CI is very small (blue square) compared to the one from the WangLUT method.

The various comparison results clearly show that the predictive abilities of the GGM are better than those of WangLUT at both the instantaneous and daily time scales, while also verifying the limitations of the simulations.

### 3.2.2. Comparison with GLASS Product

A distribution map of D-SSNR was generated with the GGM (with CI as an input) from one Landsat 5/TM scene (path 123, row 32, acquired on 01/05/2003) (TM D-SSNR, Figure 10b). Simultaneously, the GLASS SSNR was calculated in the same area (GLASS D-SSNR, Figure 10c). The original TM image in false color is also shown in Figure 10a, including different land cover types, such as urban land (light blue color), bare land (gray color), vegetation (red color), and water body (black or navy blue color).

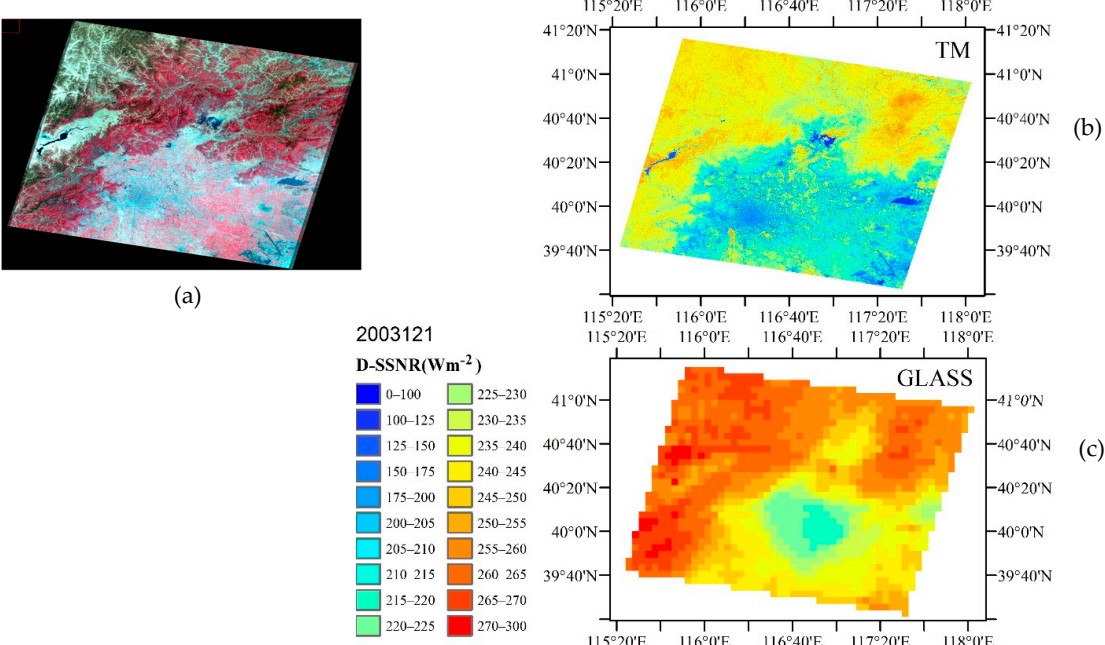

**Figure 10.** Color composite (bands 4,3,2) of the Landsat 5/TM scene (path 123, row 32) acquired on 01/05/2003 (**a**), the maps of D-SSNR from GGM (**b**) and GLASS products (**c**).

Overall, the spatial patterns of GLASS D-SSNR or TM D-SSNR are reasonable and agree well with the color composite image, e.g., the TM D-SSNR presents larger values for dense vegetation (red color in Figure 10a) because of a lower albedo, whereas for urban and bare land (blue and gray color in Figure 10a), the TM D-SSNR values are lower because of the strong reflection. Comparatively, the TM D-SSNR provides more details for characterizing the texture of different surface features and contours variations in topography (Figure 10b).

Further, the histograms of D-SSNR values from the GLASS and TM are shown in Figure 11. While the shapes are similar, the range of D-SSNR values is broader for TM than for GLASS, with a more detailed distribution, which illustrates the discrepancies caused by the coarser spatial resolution of GLASS (0.05° against 30 m for TM).

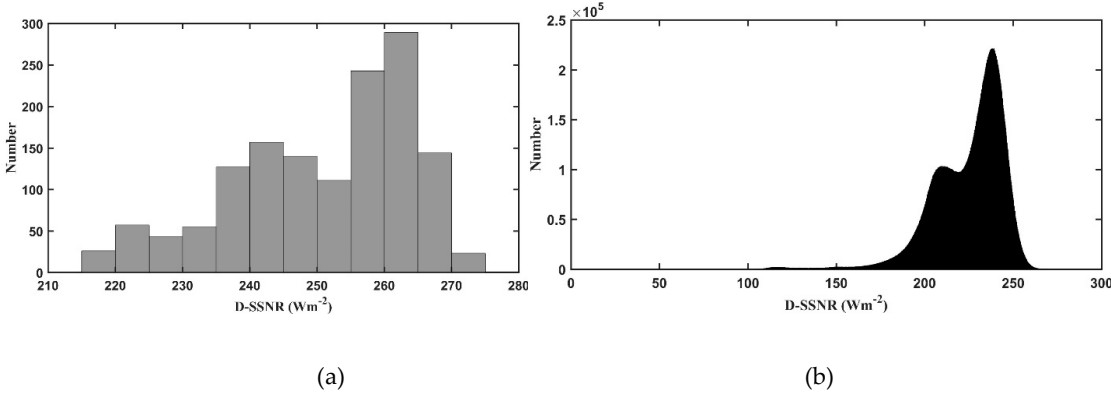

(a)　　　　　　　　　　　　　　　　　　(b)

**Figure 11.** Histograms of D-SSNR values from GLASS (**a**) and TM (**b**) for the data displayed in Figure 10b,c.

### 3.3. Model Performance Analysis

To provide a more comprehensive evaluation, the performances of the GGM for various land cover types and elevation ranges are also discussed in the following chapter, with the results of WangLUT taken as reference.

- Land cover type

As mentioned in Section 2, 11 land cover types were considered: ENF, EBF, DNF, DBF, GRA, MF, SHB, SAV, PW, CRO, and BR. The validation results for the various land cover types are shown in terms of normalized RMSE (NRMSE) to account for the different number of validation samples for each type:

$$\text{NRMSE} = \frac{RMSE}{mean(X)} \tag{5}$$

where X represents all of the observation values belonging to the same type, and the root mean square error *RMSE* is also calculated for the given type. The range of NRMSE is typically between 0 and 1, although in a few cases, such as when the values of X are small, it may exceed 1. A value of NRMSE close to 0 means that the accuracy of the estimation is high.

As shown in Figure 12, the NRMSE values from GGM are found to be generally smaller than the ones from WangLUT method for most land cover types, except in D-SSNR for EBF, and in Ins-SSNR for DNF. The estimation accuracy was found to be better for D-SSNR than for Ins-SSNR for both methods, and overall it seems that both models perform well for land cover types with low heterogeneity, such as CRO and SAV. The less good performance of the GGM for the EBF and DNF land cover types might be explained by the lesser number of samples collected for modeling. Overall, the results demonstrate the good applicability of the GGM for various land surfaces at both the instantaneous and daily scales.

- Elevation

For high-resolution images, such as those from Landsat, elevation is an important factor affecting SSNR [50]. Here, validation samples are divided into eight elevation groups, shown in Table 9, and NRMSE results for the various groups are shown in Figure 13 for both the GGM and the WangLUT method.

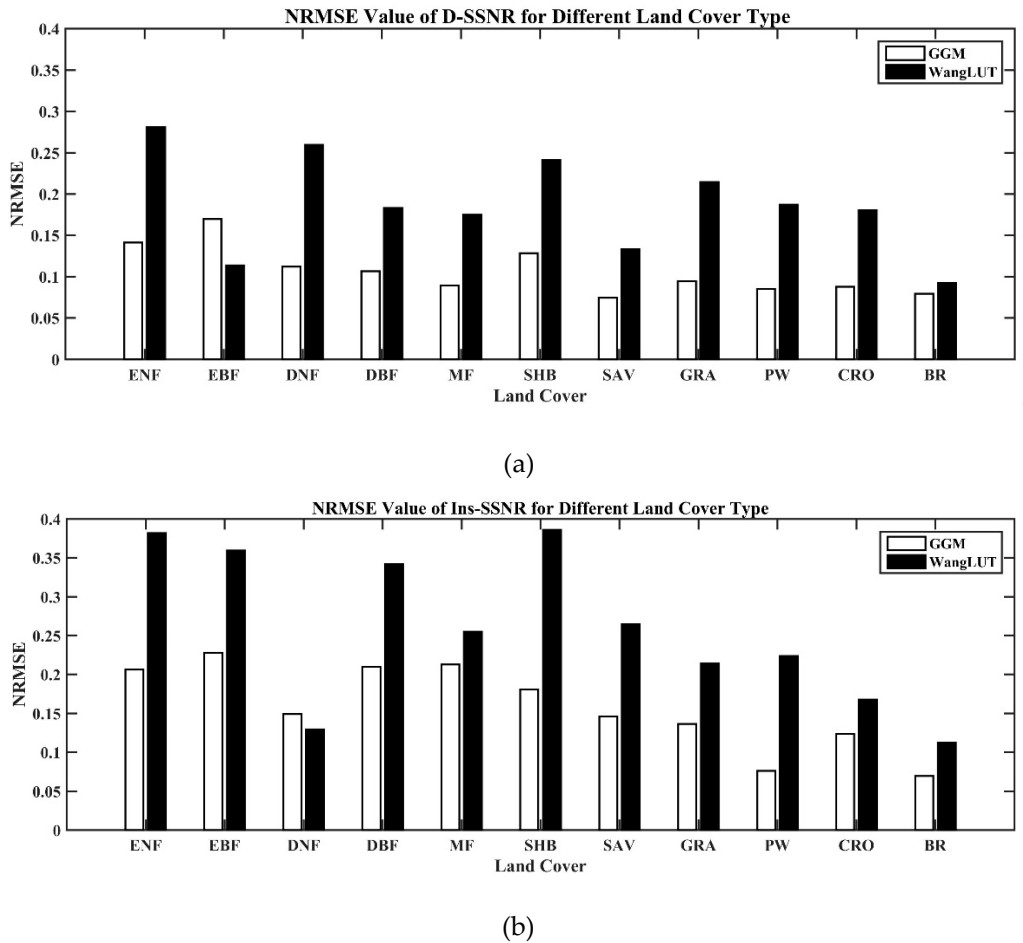

**Figure 12.** The normalized RMSE (NRMSE) of (**a**) D-SSNR and (**b**) Ins-SSNR from the GGM and the WangLUT method for different land cover types.

As shown in Figure 13, the NRMSE values from GGM are generally smaller than those from WangLUT for both D-SSNR and Ins-SSNR. The GGM values are relatively invariant and rarely exceed 0.2, except for the >3000 m interval. This may be explained by the large values of SZAs for the SSNR observations above 3000 m, which are greater than 60 degrees, and make it difficult to estimate SSNR [32]. Besides, the performances of GGM for D-SSNR estimates are better than Ins-SSNR for all elevation ranges. Note that it was inevitable that many stations are located in mountainous areas, and the performances of these methods in these regions need to be further evaluated.

• Sky condition

The applicability of GGM under various daily atmospheric conditions was also analyzed. As mentioned above, the daily average atmospheric conditions are represented by CI, and the number of validation samples in each CI interval (with a 0.1 increment) was counted and shown in Figure 14. Most observations were collected when CI was between 0.5–0.75, which means that cloudy-sky conditions were the most common. The RMSE and the NRMSE were calculated for each CI interval and are shown in Figure 15.

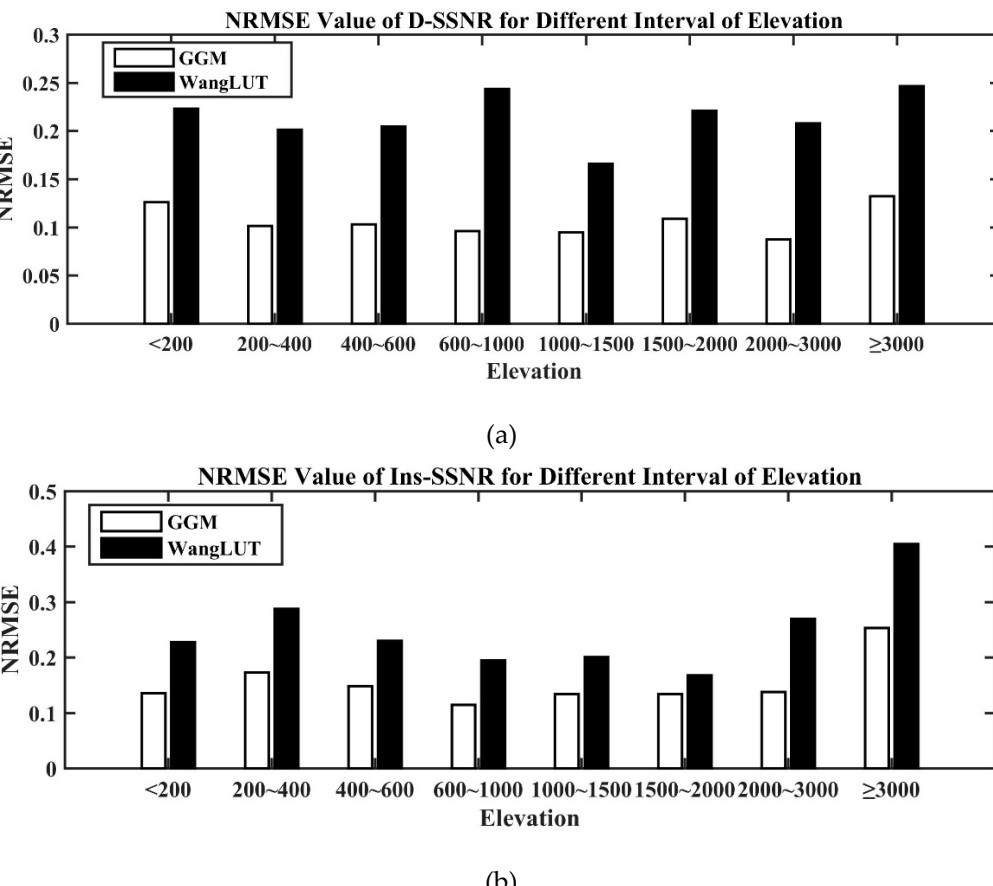

(a)

(b)

**Figure 13.** The same as Figure 12, but for different elevation ranges.

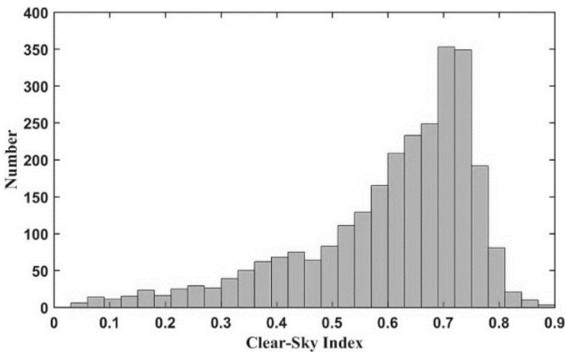

**Figure 14.** Distribution histogram of the validation dataset per CI intervals.

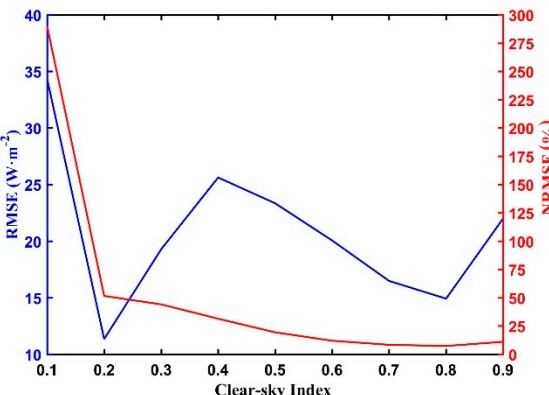

**Figure 15.** Variations in RMSE (blue line) and NRMSE (red line) of GGM per CI interval.

The NRMSE is very large (up to nearly 300%) for CI values between 0 and 0.2, indicating that D-SSNR cannot be estimated properly under overcast sky. When CI increases, the NRMSE value is gradually reduced and reaches a relatively stable value (less than 0.2). Hence, it can be concluded that the GGM works better under medium to high atmospheric transmittance conditions (CI > 0.4).

● Seasonal analysis

For better understanding the performance of GGM in different season, the variations in the validation accuracies for Ins- and D-SSNR were explored for each month shown in Figure 16. And Figure 17 gives the distribution of the validation datasets in each month at instantaneous and daily scales. It can be found that the samples were mostly collected from May to September.

As shown in Figure 17, the obvious seasonal characteristics were revealed for both Ins- and D-SSNR estimations. The NRMSE values of Ins-SSNR and D-SSNR were generally both lower from April to August than other months, and the RMSE or NRMSE variations were more stable for D-SSNR than Ins-SSNR, which was possibly caused by the intense interactions between land surface and atmosphere during this period. Overall, the variation trends is similar to the existing studies that the RMSE value is higher in summer (represents June, July, and August) than in winter (represents December, January, and February) [74], and the performances of GGM were acceptable for different seasons.

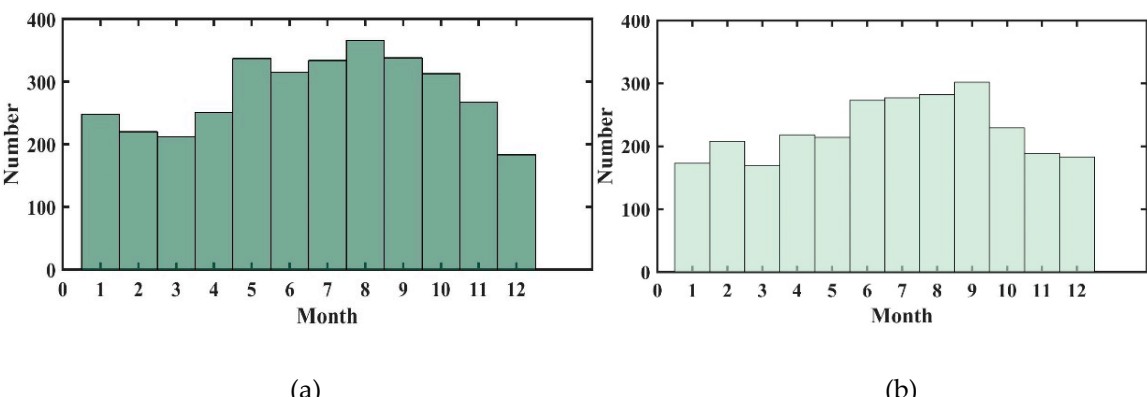

(a)                                                (b)

**Figure 16.** The histograms of the validation dataset in each month at (**a**) instantaneous and (**b**) daily scales.

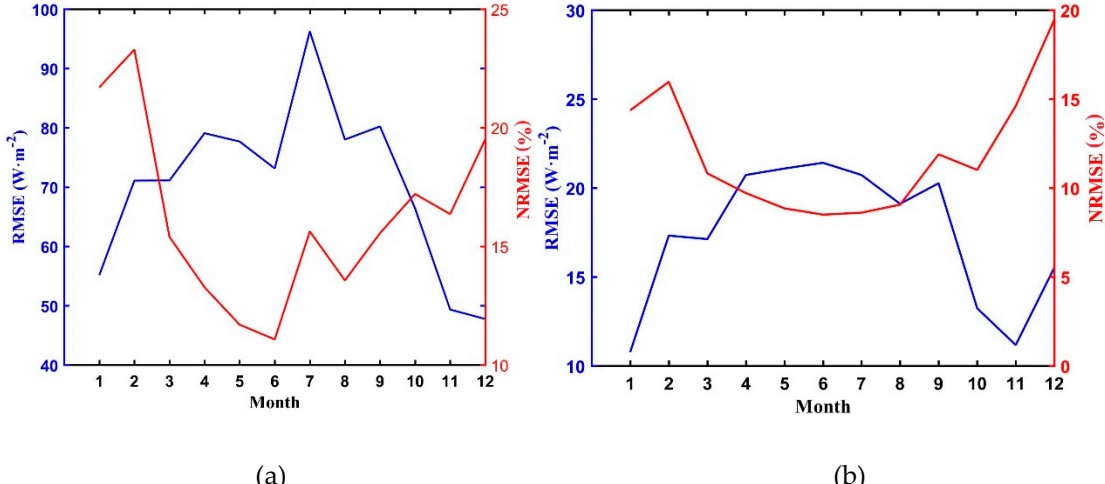

(a)　　　　　　　　　　　　　　　　　　　　　(b)

**Figure 17.** Variations in RMSE (blue line) and NRMSE (red line) of GGM per month for (**a**) Ins-SSNR and (**b**) D-SSNR.

### 3.4. Extended Use to Landsat 7/ETM+ Data

Landsat 7/ETM+ data is the subsequent version of Landsat 5/TM and the bands from the two sensors are nearly the same in the band range 1–7 (see Table 4). Hence, the GGM was directly applied to Landsat 7/ETM+ data (bands 1–7) for estimation of Ins-SSNR and D-SSNR, and the results were validated against field measurements. The validation measurements were collected from 12 sites via ARM, and the information is summarized in Table 10.

**Table 10.** Information about the 12 ARM validation sites.

| Site | Lat, Lon | Land Cover | Height (m) |
|---|---|---|---|
| Larned, Kansas: E01 | 38.20°N, 99.32°W | Cropland | 632 |
| Hillsboro, Kansas: E02 | 38.31°N, 97.30°W | Grassland | 450 |
| LeRoy, Kansas: E03 | 38.20°N, 95.60°W | Cropland | 338 |
| Plevna, Kansas: E04 | 37.95°N, 98.33°W | Rangeland | 513 |
| Halstead, Kansas: E05 | 38.11°N, 97.51°W | Wheat | 440 |
| Towanda, Kansas: E06 | 37.84°N, 97.02°W | Alfalfa | 409 |
| Elk Falls, Kansas: E07 | 37.38°N, 96.18°W | Pasture | 283 |
| Coldwater, Kansas: E08 | 37.33°N, 99.31°W | Rangeland | 664 |
| Ashton, Kansas: E09 | 37.13°N, 97.27°W | Grassland | 386 |
| Tyro, Kansas: E10 | 37.07°N, 95.79°W | Alfalfa | 248 |
| Byron, Oklahoma: E11 | 36.88°N, 98.29°W | Alfalfa | 360 |
| Pawhuska, Oklahoma: E12 | 36.84°N, 96.43°W | Prairie | 331 |

Figure 18 which shows the validation results of the Ins-SSNR and D-SSNR estimates from GGM with Landsat 7/ETM+ data, suggests that these results are satisfactory. The Ins-SSNR estimates under the all-sky condition have an overall $R^2$ value of 0.91, RMSE of 52.82 W·m$^{-2}$, MAE of 35.88 W·m$^{-2}$, and bias of –6.00 W·m$^{-2}$, while these values for the D-SSNR estimates were 0.94, 18.27 W·m$^{-2}$, 12.66 W·m$^{-2}$, and 6.03 W·m$^{-2}$, respectively.

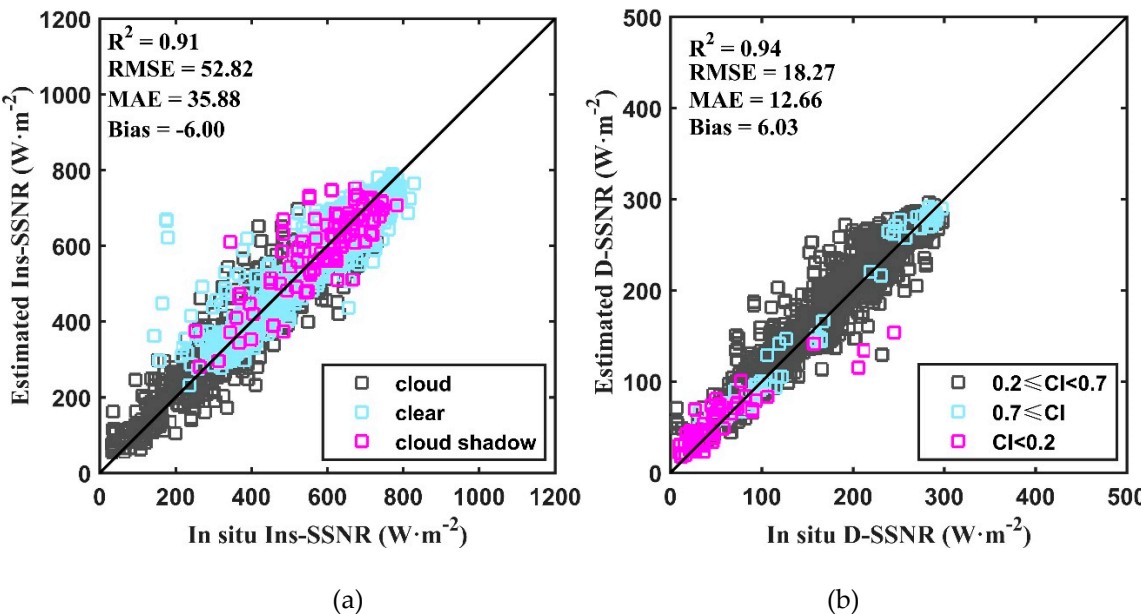

(a)                                                                    (b)

**Figure 18.** Scatterplots of the validation results of the Ins-SSNR (**a**) and D-SSNR (**b**) estimation from Landsat 7/ETM+ data.

Figure 19 displays the time-series profile and scatter plot of D-SSNR estimation with Landsat 7/ETM+ data and the in-situ measurements at the site ARM_E01 (38.202 N, 99.316 W, land cover: CRO) in the year of 2000. The GGM estimates match the in-situ observations very well and show similar patterns of variability. These results demonstrate that the GGM is also appropriate for use with Landsat 7/ETM+ data without any alteration, which will be very helpful to generate long time series of SSNR in the future.

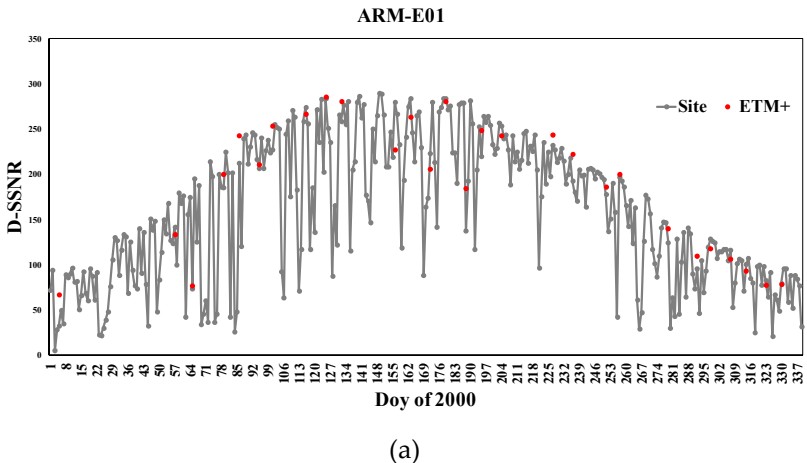

(a)

**Figure 19.** *Cont.*

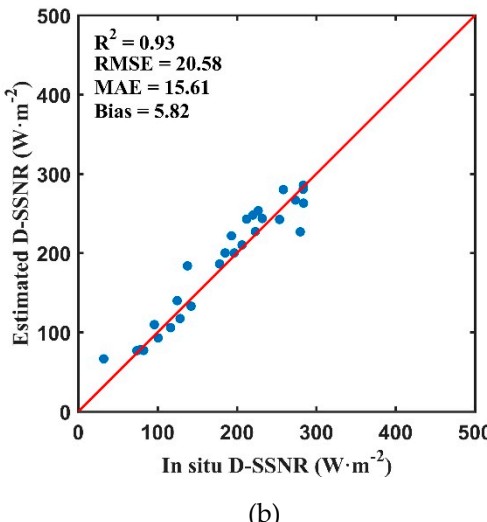

(b)

**Figure 19.** (**a**) Temporal profiles and (**b**) scatter plot of D-SSNR estimates with Landsat 7/ETM+ data and the in-situ measurements at the site ARM_E01 (38.202 N, 99.316 W, land cover: CRO) for the year 2000.

## 4. Conclusions

SSNR products at intermediate-to-high spatial resolution are required for various scientific studies and applications. Landsat 30-m satellite data provides a practical way for mapping SSNR at a relatively high-spatial resolution. Based on previous studies, this paper presents new robust empirical models to estimate the SSNR directly from Landsat TOA data and other ancillary information at both the instantaneous and daily scales. Observations obtained from 171 sites belonging to 10 global measuring networks during 1994–2011 were used for modeling and validation. These sites are distributed around the world and represent the most comprehensive information for such analysis (i.e., land cover types, climatic information, elevation, and so on). Four machine learning algorithms (MARS, BPNN, SVR, and GBRT) with different combinations of inputs using the complete set of measurements (global model) or several subsets based on the cloud mask or CI (conditional model) were developed for Ins-SSNR and D-SSNR, separately, following which the 2-fold cross-validation method was used for model evaluation, before a final determination of the optimal model properties. The results from a modified hybrid method named WangLUT and from the GLASS product were compared, and the performances of the new model were analyzed. At the end, the extended applicability of the model for Landsat 7/ETM+ data was also evaluated.

Based on the results, the GBRT global model (GGM) with BI and WV as inputs for Ins-SSNR estimates, and with CI as input for D-SSNR estimates yielded the best results. The direct validation of the estimated Ins-SSNR and D-SSNR using Landsat 5/TM data from the GGM yielded a coefficient of determination ($R^2$) value of 0.88 and 0.94, an average RMSE of 73.23 W·m$^{-2}$ (15.09%) and 18.76 W·m$^{-2}$ (11.2%), an MAE of 49.19 and 13.06 W·m$^{-2}$, and a bias of 0.64 W·m$^{-2}$ and –1.74 W·m$^{-2}$, respectively, which were generally better than the results from the WangLUT method. The analysis of the importance of each input in the GGM proved that the Ins-SSNR or D-SSNR estimate were sensitive to the TM band 1, BT, water vapor, and the CI. In particular, the addition of CI in D-SSNR estimation made the model yield more reasonable results, and improved the estimation accuracy significantly. Inter-comparison with the GLASS revealed that the TM D-SSNR display a very similar spatial distribution but with more details. Besides, the GGM is also appropriate for Landsat 7/ETM+ data without any alteration. Further analysis on the effects of land cover type, elevation, sky condition and different seasons demonstrated that the GGM has a wide applicability.

However, some limitations of the GGM exist. First, the GGM is an empirical model whose accuracy is largely determined by the variety and quality of the collected measurements. Second,

the key parameters and architecture of the GBRT algorithm were set empirically. Third, the influence of topography was not taken into account in this study. Besides, the ancillary information used (i.e., CI and WV) have a coarse spatial resolution, and may not match perfectly the site radiation observations. Moreover, Landsat 7 and 5 have a 16-day repeat cycle and provide only one observation per day; thus, the GGM is only for establishing one piece of a continuous time series of SSNR which may cause limitation of sequence analysis. All of these aspects need to be addressed in the future.

**Author Contributions:** Conceptualization, B.J. and S.L.; data curation, Y.W., Q.W., X.Z., and J.X.; investigation, Y.W.; methodology, Y.W.; resources, S.L., D.W., and T.H; validation, Y.W.; writing—original draft, Y.W.; writing—review and editing, B.J. and S.L.

**Funding:** This study was funded partly by the National Key Research and Development Program of China (2016YFA0600101), the National Natural Science Foundation of China (4197191), and the National Basic Research Program of China (973 Program): 2015CB953701.

**Acknowledgments:** This work used eddy covariance data acquired by the FLUXNET community, and in particular, by the following networks: AmeriFlux (U.S. Department of Energy, Biological and Environmental Research, Terrestrial Carbon Program (DE-FG02- 04ER63917)), AfriFlux, AsiaFlux, CarboAfrica, CarboEuropeIP, CarboItaly, CarboMont, ChinaFlux, Fluxnet- Canada (supported by CFCAS, NSERC, BIOCAP, Environment Canada, and NRCan), GreenGrass, KoFlux, LBA, NECC, OzFlux, TCOS-Siberia, USCCC. The authors acknowledge the financial support to the eddy covariance data harmonization provided by CarboEuropeIP, FAO-GTOS-TCO, Ileaps, Max Planck Institute for Biogeochemistry, National Science Foundation, University of Tuscia, Université Laval, Environment Canada and US Department of Energy and the datasets development and technical support from Berkeley Water Center, Lawrence Berkeley National Laboratory, Microsoft Researche Science, Oak Ridge National Laboratory, University of California – Berkeley and the University of Virginia. The authors would also like to thank other radiation measurements providers (listed in Table 2), and they extend their thanks to three anonymous reviewers for their valuable comments and suggestions that have greatly improved the presentation of this paper. The authors thank International Science Editing (http://www.internationalscienceediting.com ) for editing this manuscript.

**Conflicts of Interest:** The authors declare no conflict of interest.

**Appendix A**

Description of the four empirical models used for SSNR estimation.

• Multivariate Adaptive Regression Splines (MARS)

MARS is an implementation of techniques introduced by Friedman [68] for solving non-parametric regression-type problems. In this scheme, making assumptions about the underlying functional relationship between the dependent and independent variables is unnecessary, which allows it to be more flexible for modeling relationships of high dimensional data. MARS is a generalization of the stepwise linear regression procedure for fitting an adaptive nonlinear regression to data and uses expansions in piecewise linear basis functions of the form:

$$(x-t)_+ = \begin{cases} x-t & x > t \\ 0 & x \leq t \end{cases} \tag{A1}$$

and

$$(x-t)_- = \begin{cases} t-x & x < t \\ 0 & x \geq t \end{cases} \tag{A2}$$

with x=t being a knot (linear splines). The smoothing function f is a linear expansion of the basic functions:

$$f(x) = \sum_{n=1} \theta_n h_n(x_n) \tag{A3}$$

where $h_n(x_n)$ are the piecewise linear basis functions and $\theta_n$ are the coefficients that are estimated by minimizing the residual sum-of-squares using standard linear regression.

In this study, MARS was first applied for Ins-SSNR and D-SSNR estimation. It was implemented on the R platform with the package "earth" [75], in which the input variables can be selected automatically. To obtain the optimal MARS model, the grid search method [68] was applied to determine the best parameter set of degree so that the minimum forecasting RMSE was generated. The backward stepwise process was carried out to train the MARS model.

● Support Vector Regression (SVR)

The SVR is a regression method using the same principles as the support vector machine (SVM) for classification, with only a few minor differences. It is proposed by [70], and although less popular than SVM, SVR has been proven to be an effective tool in real-value function estimation. As a supervised-learning approach, SVR trains using a symmetrical loss function, which equally penalizes high and low misestimates. Using Vapnik's-insensitive approach, a flexible tube of minimal radius is formed symmetrically around the estimated function, so that the absolute values of errors less than a certain threshold are ignored both above and below the estimate [76]. In this manner, points outside the tube are penalized, while those within the tube, either above or below the function, receive no penalty. One of the main advantages of SVR is that its computational complexity does not depend on the dimensionality of the input space. Additionally, it has excellent generalization capability, with high prediction accuracy. More detail about SVR can be found in [77].

In this study, SVR was implemented on the R platform with the package "kernlab" [78]. The kernel function "rbfdot" (Radial Basis kernel "Gaussian") was used in training and predicting. After looping in two parameters (C and sigma) threshold, the optimal parameterization scheme for different conditions were finally determined through the evaluation results (highest $R^2$ value and lowest MBE and RMSE values).

● Backpropagation Neural Network (BPNN)

BPNN was used as an empirical nonlinear statistical method in a variety of applications and is the most widely applied neural network [69,70]. BPNN has proven to be an effective algorithm for estimating surface radiation budget variables, such as the incident shortwave radiation [71]. Therefore, BPNN was selected for the comparison of the performance among other methods in this study. A BPNN is a collection of electrical neurons interconnected to each other in various topologies and can be applied to any model in which the output variables are computed from the input variables. Figure A1 shows the architecture of the BPNN model used in this study, consisting of four layers of neurons: Input layer, two hidden layers, and output layer. Input $\{x_j\}_{j=1}^{m}$ is transmitted through a connection that multiplies its strength by a weight represented by $\{w_{ij}\}_{i=1}^{k}$. This gives the value $x_i w_{ij}$, which is an argument to a transfer function f that yields an output $y_i$.

$$y_i = f\left(\sum_{j=1}^{m} w_{ij} x_j\right) \tag{A4}$$

where *i* is the index of neuron in the hidden layer and *j* is the index of inputs to the neural network.

The BPNN undergoes an adaptation cycle, during which the weights are updated until the network reaches a state of equilibrium. The BPNN activation function in the hidden layer was set to "logsig", the transfer function for the output layer to "purelin", and the training function to "trainlm", respectively.

In this study, the SSNR estimation utilized MATLAB Neural Net Tool Kit with the Matlab R2015b platform (The MathWorks, Natick, 2015). A typical feed-forward network trained with a resilient backpropagation algorithm [79,80] is employed in modeling. Finally, the optimal number of nodes in the hidden layers for different conditions were finally determined empirically.

● Gradient Boosting Regression Tree

The GBRT is a flexible non-parametric statistical tool, firstly proposed by Friedman [71], that can be used in complex classification and prediction without pre-specifying the type of the relationship between input and response variables. Among algorithmic models, GBRT is unique in the sense that it achieves better predictive capacity than a single decision tree with large data size. The core idea of this model consists in producing a prediction model by constructing an M amount of different weak classifiers through multiple iterations in order to produce a highly accurate prediction rule. Each iteration is designed to improve the previous result by reducing the residuals of the previous model and establishing a new combination model in the gradient direction of the residual reduction. The core idea of GBRT gives it a natural advantage to discover a variety of distinguishing features and feature combinations. The general process of GBRT are shown in Yang et al. [81], and additional details of GBRT can be found in Hastie et al. [82] and Ridgeway [83].

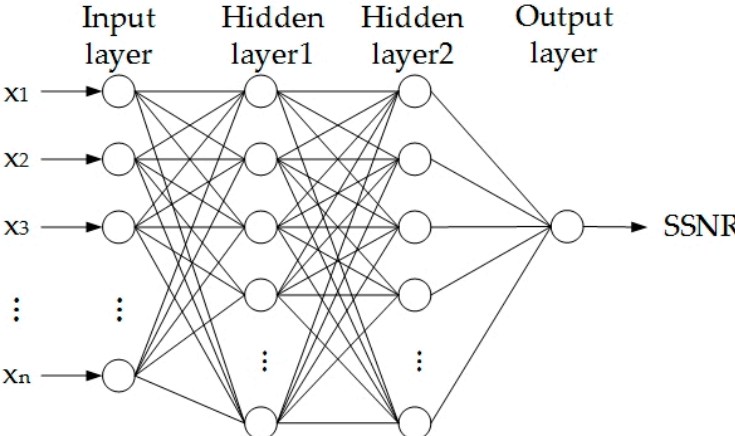

**Figure A1.** Backpropagation neural network (BPNN) with multi-input-one-output architecture.

In this study, GBRT was implemented on the R platform with the package "xgboost" [84]. After looping in four parameters ("nround", "max_depth", "eta", "subsample", and "min_child_weight"), the threshold for the optimal parameter set was finally determined.

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
