# Peer review of "Surface Shortwave Net Radiation Estimation from Landsat TM/ETM+ Data Using Four Machine Learning Algorithms"

_remotesensing, doi:10.3390/rs11232847_

Round 1

Reviewer 1 Report

Authors conducted this study with the objective of uncovering an optimal machine-learning method to evaluate Nst Surface Radiation product. Several key recent reference citations have been missed:

1. Jia, A.L.; Liang, S.L.; Jiang, B.; Zhang, X.T.; Wang, G.X. Comprehensive Assessment of Global Surface Net Radiation Products and Uncertainty Analysis. J. Geophys. Res. Atmos. 2018, 123, 1970–1989. [CrossRef]

2. Stephens, G.L.; Li, J.L.; Wild, M.; Clayson, C.A.; Loeb, N.; Kato, S.; L’Ecuyer, T.; Stackhouse, P.W.; Lebsock, M.; Andrews, T. An update on Earth’s energy balance in light of the latest global observations. Nat. Geosci. 2012, 5, 691–696. [CrossRef]

3. Liang, S.L.; Wang, K.C.; Zhang, X.T.; Wild, M. Review on Estimation of Land Surface Radiation and Energy Budgets from Ground Measurement, Remote Sensing and Model Simulations. IEEE J. Stars 2010, 3, 225–240. [CrossRef]

4. Inamdar, A.K.; Guillevic, P.C. Net Surface Shortwave Radiation from GOES Imagery-Product Evaluation Using Ground-Based Measurements from SURFRAD. Remote Sens. 2015, 7, 10788–10814. [CrossRef]

5. Zhang, X.Y.; Li, L.L. Estimating net surface shortwave radiation from Chinese geostationary meteorological satellite FengYun-2D (FY-2D) data under clear sky. Opt. Express 2016, 24, A476–A487. [CrossRef]

6. Duan, S.B.; Li, Z.L.; Leng, P. A framework for the retrieval of all-weather land surface temperature at a high spatial resolution from polar-orbiting thermal infrared and passive microwave data. Remote Sens. Environ. 2017, 195, 107–117. [CrossRef] 8. Tarpley, J.D. Estimating Incident Solar-Radiation at the Surface from Geostationary Satellite Data. J. Appl

The above citations are all relevant to highlight the importance of this study.

Also a recent paper "Benchmarking Machine Learning Algorithms for Instantaneous Net Surface Shortwave Radiation Retrieval Using Remote Sensing Data" by Hua Wu and Wangmin Ying is a similar study and yields much better accuracy (RMS of less than 55 W m-2) than this study.

Reviewer 2 Report

This paper presents a study on the improvement of net radiation estiamtion using Landsat data and machine laerning algorithms. The paper is of interest for the solar energy community in general. The methodology and results are clearly explained, however there are two minor things to correct: 

In page 13, line 353, he GLASS should be the GLASS. In page 14, line 380, explain the definition of the different land cover types.

Reviewer 3 Report

This paper is reasonably well written and I believe it will be worthy of publication with some modest adjustments.  It has a substantial amount of information and good research methods. That being said, there are some substantial and annoying logistical problems involving figures and tables that need to be immediately remedied.   I’m going to submit this partial review, since I feel it necessary to get a second cut as soon as possible where the Figures and Tables are corrected.  I anticipate that the next corrected version will require minimal additional comments.

There are two Figure 1’s in their legends, approximately lines 103 and 127.   This means that the second Figure 1 legend should be Figure 2 and all other figure legend numbers above increased by one.   The references to figure numbers in the text are OK. Figure 5 (which needs to be changed to Figure 6 in the legend) is not referenced in the text and needs to be.

Table 4, as described at approximately line 153, does not exist and needs to be added.   It’s supposed to show the characteristics of Landsat spectral bands, but the Table 4 presented (approximately line 245) does nothing of the sort.   So then, Table 4 needs to be added and all other table legend numbers above be incremented by one.  The table numbers in the text are also correct.  It’s a little more difficult to interpret results without Table 4 giving information on the Landsat bands.

The strengths of this paper are that good results are achieved using statistical methods, a modest number of inputs and avoiding potentially laborious radiative transfer calculations.  The validation measurements are made over a large variety of conditions, adding to the generality of the methods and are at the high spatial resolution of Landsat data.  Based on the repeat cycle of Landsat measurements (approximately 16 days), however, the authors need to acknowledge that this is only one piece of establishing a continuous time series of net shortwave radiation at the surface.

Around line 159, a bit more information, please, on the data and methods that go into the estimation of the cloud mask, which turns out to be an important qualifier for the Ins results.  Presumably, it involves Landsat data plus what else?   Similarly, what conditions/data establish “cloud shadow” events, a bit more of an obscure quantity?

Around line 277, please elaborate on the “sinusoidal model”.   I presume that it involves using varying earth-sun angle values over the course of the day to get D-SSNR with a (once-daily?) input of CI?   How many events during the day are calculated to get the daily total/average. The CI already has implicit information about atmospheric clearness/transmission/cloudiness (Eq. 2) and the downwelling solar at the surface (Rg), so using it presumably provides a significant leg up when entered into the statistical estimation methods for D-SSNR, since it already contains some form of Rg information.   This seems a bit like cheating and may be responsible for the daily results being of higher quality than the instantaneous, something I would not have suspected would happen. Comments please?

The description of Figure 5 (labeled Figure 4 in the figure legend) is not adequate and the Figure itself is of poor quality.   Between the poor description and possibly poor quality (hard to tell the figure quality since it’s not obvious in the description what I should be looking for), I had a hard time interpreting this Figure.  Please clarify what this Figure is supposed to be showing.

I realize that this paper emphasizes statistical methods and it is clear why most of the model inputs have important effects on SSNR results.   However, any physical insight into why BT’s provide information to the statistics would be welcome, they are not obvious to me.   All I can think of is somehow the BT’s have implicit information on the surface type.  Comments?

As a suggestion for later work, I would substitute the cosine of the various angles for the angles themselves, if not already done.  I would also normalize the precipitable water by the cosine of the sun angle, since the water path increases as the solar path length increases.

I will be waiting for a version with Figures and Tables corrected before a last and hopefully brief review.  Good luck.

Round 2

Reviewer 1 Report

Authors have addressed the concerns satisfactorily. Only one minor comment here.

Line # 514 fix the typo error fig2020 to fig 20. The ARM SGP E01 site have 15 -min-averaged measurements but I see very few points for observations (red), may be because of the large number of points ? I suggest use the 15 min-averaged points (if not in the time series as shown in fig 20) but as a scatter plot between D-SSNR and GGM alongwiith associated error statistics (bias and RMS).
